# Microtubule-Associated Protein ATIP3, an Emerging Target for Personalized Medicine in Breast Cancer

**DOI:** 10.3390/cells10051080

**Published:** 2021-05-01

**Authors:** Maria M. Haykal, Sylvie Rodrigues-Ferreira, Clara Nahmias

**Affiliations:** 1Institut Gustave Roussy, Université Paris-Saclay, Inserm U981, Biomarqueurs Prédictifs et Nouvelles Stratégies Thérapeutiques en Oncologie, 94800 Villejuif, France; maria.haykal@gustaveroussy.fr (M.M.H.); sylvie.rodrigues-ferreira@gustaveroussy.fr (S.R.-F.); 2LERMIT Laboratory, 92296 Chatenay-Malabry, France; 3Inovarion, 75005 Paris, France

**Keywords:** *MTUS1*, tumor suppressor, breast cancer, prognostic biomarker, predictive biomarker, microtubule, taxanes, chemotherapy, targeted therapy

## Abstract

Breast cancer is the leading cause of death by malignancy among women worldwide. Clinical data and molecular characteristics of breast tumors are essential to guide clinician’s therapeutic decisions. In the new era of precision medicine, that aims at personalizing the treatment for each patient, there is urgent need to identify robust companion biomarkers for new targeted therapies. This review focuses on ATIP3, a potent anti-cancer protein encoded by candidate tumor suppressor gene *MTUS1*, whose expression levels are markedly down-regulated in breast cancer. ATIP3 is a microtubule-associated protein identified both as a prognostic biomarker of patient survival and a predictive biomarker of breast tumors response to taxane-based chemotherapy. We present here recent studies pointing out ATIP3 as an emerging anti-cancer protein and a potential companion biomarker to be combined with future personalized therapy against ATIP3-deficient breast cancer.

## 1. ATIP3 and the *MTUS1* Gene, a Historical Point of View

The microtubule-associated tumor suppressor (*MTUS1*) gene was first identified in 2003 under the name *MTSG1* [1]. This gene is located at chromosomal position 8p22, a region frequently reported to be lost in a number of solid tumors, including breast cancer [2,3]. In their search for new tumor suppressor genes, Seibold and collaborators used a differential display RT-PCR strategy and identified the *MTSG1* transcript as being up-regulated in 3-dimensional cultures of quiescent versus differentiated human endothelial cells [1]. *MTSG1* was reported to encode a 436 amino-acids polypeptide (48 KDa) co-localizing with mitochondria, that was down-regulated in pancreatic cancer and inhibited cell proliferation when expressed into pancreatic cancer cells.

In an independent study published a few months later by our group, the same polypeptide was identified in a yeast two-hybrid system as an intracellular interacting partner of the human angiotensin II AT2 receptor and was designated ATIP1 [4]. AT2 is a rare example of a seven transmembrane receptor that controls cell proliferation through intracellular pathways that do not use typical G-protein signaling [5]. ATIP1 was, thus, identified as a scaffold protein mediating the anti-proliferative effects of AT2 in a constitutive fashion, even in the absence of receptor stimulation [4].

The functional relevance of ATIP1-AT2 receptor complexes was further demonstrated in a number of cell types of cardiovascular, adipose, and neuronal origins [6,7,8,9,10,11,12]. Transgenic animals overexpressing ATIP1 were useful for demonstrating the role of the AT2/ATIP1 axis in pathophysiological models of neointima formation [6,7], vascular senescence [8] and endothelial dysfunction [9], as well as neuronal differentiation [10] and adipose tissue inflammation [11]. The observation that ATIP1 and AT2 transcripts are co-regulated by PARP-1 [12] further supported the tight link between these two proteins. ATIP1 was also described as a Golgi-associated protein involved in intracellular trafficking of the AT2 receptor to the cell membrane [13]. Thus, ATIP1 mainly appears as a regulator of AT2 receptor functions in cardiovascular and central nervous systems.

ATIP1 belongs to the family of evolutionary conserved AT2-interacting proteins (ATIP), that includes ATIP3 and ATIP4. All ATIPs share the same C-terminal amino-acid sequence of 396 residues comprising the AT2 receptor binding site and several coiled-coil motifs involved in homo- and hetero-dimerization [4]. ATIP1, ATIP3, and ATIP4 are the products of alternative splicing and different promoter usage of the same *MTUS1* gene, organized into 17 coding exons [14,15]. *MTUS1* encodes two splice variants of ATIP3, designated ATIP3a and ATIP3b, that differ by a single in-phase exon encoding a sequence of 60 amino-acids in the N-terminal portion of the proteins. To date, no functional difference has been reported between the two polypeptides. In rodents, the ortholog *MTUS1* gene comprises 15 coding exons that are also alternatively spliced [13,16] to generate 3 different ATIP isoforms (designated ATBP50, ATBP135, and ATBP60 in the mouse) with high sequence homology to human ATIP1, ATIP3 and ATIP4, respectively. The *Xenopus MTUS1* gene ortholog encodes the ICIS protein, which exhibits structural homology with the mammalian ATIP3 isoform [14].

The ATIP1 and ATIP3 transcripts display an ubiquitous profile with high expression in the brain, ATIP3 being the prominent isoform in peripheral tissues, whereas ATIP4 is exclusively expressed in the central nervous system [14]. The presence of a canonical transmembrane domain in its polypeptide sequence suggests close interaction of ATIP4 with the AT2 receptor at the plasma membrane. However, this isoform has never been characterized at the molecular nor functional level.

Consistent with the initial report that *MTUS1* may be a candidate tumor suppressor gene in pancreatic cancer [1], inactivation of the gene in knock-out animals was associated with B cell lymphoproliferative disease [17]. Furthermore, p53-regulation of ATIP1 transcripts suggested a link between *MTUS1* gene regulation and cancer [18]. Indeed, *MTUS1* down-regulation in cancer tissues was frequently reported, including in tumors from the breast [19,20,21,22,23], bladder [24,25], colon [26,27,28,29], gallbladder [30], gastric tissues [31,32], lung (NSCLC) [33], head-and-neck [34,35,36,37,38,39], clear cell renal cell carcinoma (cc-RCC) [40,41], and uveal melanoma [42], with the exception of prostate cancer, in which *MTUS1* expression was reported to increase with cancer progression [43,44] (Table 1). Only few studies were designed to discriminate between different ATIP isoforms, and they all pointed to ATIP3 as the major *MTUS1* isoform altered in human malignancies [19,21,36,43], ATIP1 being a minor form expressed in normal peripheral tissues [14].

The present review focuses on the characterization of ATIP3 in breast cancer. We summarize recent results investigating intracellular mechanisms regulated by this protein and we present evidence that ATIP3 is a prognostic and predictive biomarker in breast tumors. Finally, we discuss data suggesting that ATIP3 studies may open the way to important emerging targets for anti-cancer therapy.

## 2. ATIP3 Is a Microtubule-Associated Protein

ATIP3 is a 1270 amino-acids polypeptide organized into an unstructured N-terminal region of 874 amino-acids and a coiled-coil C-terminal region shared with other ATIP members [14,20]. Initial analyses of its intracellular localization clearly indicated that ATIP3 decorates the microtubule cytoskeleton and the centrosome in interphase, and localizes at the mitotic spindle during all stages of mitosis. In microtubule co-sedimentation assays, ATIP3 was found to associate with stable microtubules rather than soluble tubulin [19]. Studies of ATIP3 deletion mutants revealed that ATIP3 binds microtubules through a positively charged, central region (D2) of the protein [21]. These basic residues are believed to interact with the acidic charges of tubulin tails, as reported for other microtubule-associated proteins (MAPs).

The microtubule cytoskeleton plays an essential role in cell homeostasis by controlling not only cell shape, but also intracellular trafficking of proteins and organelles, as well as cell migration and mitosis. Microtubules are polarized and very dynamic structures formed by the assembly of a/b tubulin dimers at their growing (plus) ends, in a GTP-dependent manner. Microtubule ends are constantly alternating between phases of growth (polymerization) and shrinkage (depolymerization), in a process known as “dynamic instability” [45]. This process is essential to allow rapid adaptation of the cytoskeleton to cell changes, such as formation of the mitotic spindle and response to extracellular cues. Microtubule assembly and dynamics are tightly regulated by a large number of MAPs, including structural MAPs that localize along the microtubule fibers, and microtubule plus ends-tracking proteins (+TIPs) that decorate the rapidly growing plus ends [46]. Among +TIPs, End-Binding proteins EB1 and EB3 play a central role in regulating microtubule dynamics. EB1 directly binds the microtubule plus ends [47], where it acts as a platform to recruit many other regulatory +TIPs [46]. Furthermore, EB1 binding by itself was shown to accelerate the maturation of microtubule plus ends, thereby contributing to dynamic instability [48].

ATIP3 is a structural MAP localized all along the microtubule lattice, and a potent microtubule stabilizer [19]. Strikingly, ATIP3 reduces microtubule dynamics at plus ends although it does not bind to this location [21]. ATIP3 was actually found to interact with EB1 in the cytosol and reduce free EB1 turnover on its preferential site at the plus ends [49]. In this regard, ATIP3 may be considered as an “endogenous antagonist” of EB1, like other structural MAPs, such as MAP1B, MAP2, or MAP-Tau, that all interact with EB1 to restrain its accumulation at growing ends [50]. By preventing EB1 accumulation at plus ends, cytosolic ATIP3/EB1 complexes control the rate of microtubule growth and shrinkage, thereby regulating microtubule targeting to the cell cortex, and subsequent cell polarity and migration [49] (Figure 1). In the absence of ATIP3, EB1 is free to accumulate on plus ends, which accelerates microtubule dynamics. This in turn increases cancer cell motility, in agreement with increased metastatic behavior of ATIP3-deficient breast tumors.

## 3. Cancer-Related Molecular Mechanisms Controlled by ATIP3

The microtubule stabilizing properties of ATIP3 are consistent with its potent anti-cancer effects. Indeed, ectopic expression of ATIP3 in breast cancer cells was shown to markedly reduce tumor growth [19] and distant metastasis [21] in pre-clinical models. In line with these in vivo studies, ATIP3 expression reduces cell proliferation, prolongs the time spent in mitosis and reduces cell polarity and migration. However, the intracellular mechanisms associated with the anti-cancer effects of ATIP3 have only recently emerged.

A major step towards the understanding of ATIP3-associated molecular mechanisms was recently provided by a proteomic approach that aimed at identifying intracellular interacting partners of ATIP3 in breast cancer cells. Co-immunoprecipitation experiments followed by mass spectrometry led to the identification of 145 ATIP3-interacting proteins, among which, nine were related to the microtubule cytoskeleton and/or mitosis [51]. Interestingly, ATIP3 interacts with KIF2A—a microtubule depolymerizing kinesin of the KinI family—and its regulator, Dda3, via a minimal sequence of 112 amino-acids present in the central basic region of the protein. The ATIP3/KIF2A/Dda3 complex prevents the accumulation of KIF2A at the poles of the mitotic spindle, and therefore controls microtubule depolymerization and spindle dynamics at minus ends (Figure 1). As a consequence, ATIP3 regulates the microtubule poleward flux—a mechanism of concerted polymerization at plus ends and depolymerization at minus ends of the spindle—that takes place in metaphase to maintain a constant size of the mitotic spindle [52]. In ATIP3-depleted cells, the mitotic spindle is significantly shortened [51]. This mitotic abnormality, among others, is expected to provoke major defects in chromosome segregation and subsequent aneuploidy.

The stability of the ATIP3/KIF2A/Dda3 molecular complex requires phosphorylation by the mitotic kinase, Aurora A, a major kinase deregulated in cancer, including breast cancer [53,54]. Aurora kinase A is known to localize at the spindle poles where it phosphorylates KIF2A to reduce both the amount, and the depolymerizing activity, of the kinesin at this location. Of importance, ATIP3 was shown to maintain an active pool of Aurora kinase A at the poles, as a mechanism to control KIF2A activity and mitotic spindle integrity [51] (Figure 1). In an independent study conducted in renal cancer cells, ATIP3 was found to regulate the phosphorylation of KIF2C (also designated MCAK), another microtubule depolymerizing kinesin of the KinI family. A recombinant fragment of ATIP3 was shown to contribute to KIF2C phosphorylation on serine 192 by Aurora kinase B, and increase tubulin polymerization, consistent with its microtubule stabilizing effects [40].

Interestingly, the ATIP3 ortholog in *Xenopus*, named ICIS, is also a microtubule-associated protein that interacts with XKCM1 (the *Xenopus* ortholog of MCAK) and with Aurora B kinase to control the integrity of the mitotic spindle [55]. However, in contrast to human ATIP3, ICIS does not decorate the mitotic spindle but localizes both at centromeres and centrosomes in mitotic cells. Furthermore, ICIS stimulates, rather than inhibits, the depolymerizing activity of MCAK to control spindle dynamics at the kinetochores and chromosome segregation in anaphase. Other studies [56] have shown that in *Xenopus* mitotic extracts, ICIS interacts with both MCAK and KIF2A in addition to Aurora B, INCENP, and TD-60, all members of the chromosome passenger complex (CPC, a master regulator of faithful mitosis), supporting the notion that ICIS functions as a scaffold located at the inner centromere to regulate microtubule depolymerization and dynamics at the kinetochore. Together, these studies raise a common scenario for ATIP3 mechanisms of action on mitotic spindle integrity and chromosome segregation in different cellular models. Depending on the organism and cell type, ATIP3 interacts with different kinesins of the KinI family (either KIF2A and/or MCAK) to regulate their depolymerizing activity through Aurora (A or B) kinase-dependent phosphorylation, with a major effect on microtubule dynamics and mitosis (Table 2).

Besides its prominent effects on microtubule dynamics through interaction with EB1, KinI kinesins, and Aurora kinases, it is likely that ATIP3 may control other molecular mechanisms that may account for its potent anti-cancer and anti-metastatic effects in different malignancies. In salivary adenoid cystic carcinoma [36] and squamous carcinoma of the tongue [37], ATIP3 was found to inhibit the phosphorylation of extracellular regulated kinases ERK1/2, as well as the expression of epithelial to mesenchymal (EMT) markers slug and vimentin. In ovarian cancer cell lines, anti-migratory and anti-metastatic effects of ATIP3 were also related to inhibition of the ERK/EMT axis [57]. More recently, ATIP3 expression has been associated with reduced ERK1/2 phosphorylation, cell proliferation, and migration in gastric cancer [32]. In this model, ATIP3 was also found to inhibit the activity of CDC25B phosphatase, leading to phosphorylation and inhibition of the master cell cycle kinase CDK1 [32]. Interestingly, previous studies have revealed that expression of the ATIP1 isoform reduces ERK1/2 activity induced by receptor tyrosine kinases [4], which further links the ATIP family with inhibition of the ERK pathway (Table 2). While the molecular mechanisms by which ATIP3 inhibits ERK1/2 phosphorylation remain to be clarified, these findings clearly warrant further investigation in breast cancer.

## 4. ATIP3 Is a Prognostic Biomarker in Breast Cancer

Breast cancer is the leading cause of death by malignancy in women all over the world. Major difficulties faced by clinicians in treating their patients are related to the high heterogeneity of breast tumors and their ability to metastasize to distant organs. At the onset of the 21st century, the classification of breast tumors into distinct molecular subtypes (luminal, HER2, or triple-negative breast cancer (TNBC)), based on the expression of hormone receptors for estrogen (ER) and progesterone (PR), and amplification of the HER2 oncogene, has changed the paradigm and oriented clinical decisions [59,60]. Over the past few years, the rapid development of high-throughput molecular techniques investigating genomic, epigenetic, and transcriptional alterations in breast tumors has launched a new area of cancer research. Precision medicine, which aims at administrating the right treatment to the right patient based on unique molecular properties of each tumor, is considered today as a major endpoint in the fight against cancer. This approach mainly relies on the identification of biomarkers to select the appropriate population of patients for personalized treatment.

With the aim of identifying new molecular markers in breast cancer, the expression levels of the *MTUS1* gene were analyzed in a DNA array study of 151 breast tumors compared to normal breast tissue [19]. These studies revealed for the first time that *MTUS1* is markedly down-regulated in approximately 50% of all breast cancers and 70% of TNBC, which represent the most aggressive tumors. Real-time RT-PCR analyses using specific oligonucleotides indicated that ATIP3 is the major transcript expressed in normal mammary gland and down-regulated in breast tumors. In several independent cohorts of patients, low levels of ATIP3 mRNA were significantly associated with TNBC subtype [61,62], high grade, and metastatic breast tumors [19], thereby linking low ATIP3 expression and breast cancer aggressiveness.

The mechanisms by which ATIP3 mRNA levels are reduced in breast cancer have not yet been clarified. It is to note that the human ATIP3 promoter contains several CpG islands, suggesting possible regulation by promoter methylation at these sites. Although this possibility has not been directly addressed in breast cancer, recent studies have indeed reported promoter methylation as a possible mechanism for *MTUS1* down-regulation in non-small cell lung (NSCLC) carcinoma [33]. Another mechanism for regulation of *MTUS1* mRNA stability by the RNA binding protein SORBS2 was recently reported in clear cell renal cell carcinoma [40]. Long non-coding RNAs were shown to control the stability of *MTUS1* transcripts via microRNAs in gastric [32] and cervical cancer [63]. In several other cancer types, including breast [64], colorectal [28], lung [65], gallbladder cancer [30], and osteosarcoma [66], microRNAs also down-regulate *MTUS1* expression. Most microRNAs were found to target the 3’UTR of the gene, which is common to all ATIP isoforms. At the genomic level, alterations of ATIP3-specific coding exons by somatic mutation have been identified in hepatocellular carcinoma [67] but mutational analysis of *MTUS1* in breast cancer remains to be performed. A single study has reported genomic deletion of a sequence corresponding to ATIP3-specific exon 4 in association with increased familial risk of breast cancer [68]. Together these studies suggest that ATIP3 alterations in cancer are a consequence of deregulated gene expression rather than genomic variations.

To investigate whether ATIP3 may represent a prognostic biomarker of breast cancer patient survival, Kaplan-Meyer curves were extracted from different cohorts of breast cancer patients. These studies indicated that low ATIP3 levels are significantly associated with poor clinical outcome and reduced 5-years survival [21]. ATIP3 was also identified as a prognostic biomarker of relapse-free survival and overall survival among patients with metastatic disease. Of interest, the *MTUS1* gene was also described as an interesting prognostic biomarker of patient clinical outcome in other cancer types (Table 1).

The increasing amount of publicly available large-scale molecular studies of breast tumors, associated with clinical data of the patients, have opened the possibility to explore the prognostic value of biomarker combinations, that are likely to be more accurate and informative than single biomarkers. In this context, the prognostic value of ATIP3 was studied in combination with its interacting partner EB1, that was shown to be up-regulated in aggressive breast tumors [69]. Functional studies mentioned above [49,50] showing that ATIP3 antagonizes the effects of EB1 on microtubule dynamics, raised the possibility that tumors with low ATIP3 and high EB1 levels may be associated with increased malignancy and worse prognosis compared with other breast tumors. Studies conducted on 5 independent cohorts of breast cancer patients indeed confirmed the increased prognostic value of combined ATIP3/EB1 expression compared with each biomarker alone [22,70].

In this regard, it will be interesting to further investigate the value of combined expression of ATIP3 with other partners involved in important molecular complexes, such as the depolymerizing kinesins (KIF2A/KIF2C) and mitotic (Aurora) kinases that have also been described as prognostic biomarkers of breast cancer patient survival [71,72,73,74]. Other MAPs have been identified as prognostic biomarkers in breast cancer [70]. They are part of protein networks that coordinately regulate microtubule dynamics and functions. Prognostic value of their combined expression warrants further examination.

## 5. ATIP3 Is a Predictive Biomarker of Taxane-Based Chemotherapy in Breast Cancer

The observation that ATIP3 is a stabilizing MAP raised the possibility that its expression may impact the effects of taxanes on cancer cells. Taxanes (paclitaxel and docetaxel) are chemotherapeutic agents widely used for the treatment of breast cancer. They are generally used in neoadjuvant settings (to reduce tumor size before surgery) and in adjuvant treatment for TNBC and metastatic breast tumors. Taxanes are also frequently used for treating other malignancies, such as ovarian, prostate, lung, and pancreatic cancers. These drugs, also called mitotic poisons, bind microtubules at the “taxane site” and block microtubule dynamics. By stabilizing spindle microtubules in mitosis, taxanes promote mitotic arrest at the spindle assembly checkpoint (SAC), which results in apoptotic cell death. At very low doses of a few nanomolar range, taxanes induce the formation of multipolar spindles and other mitotic defects leading to aneuploidy [75].

Because microtubules are essential components of all cell types, taxanes have very severe side effects characterized by neuropathies and immune system defects, which strongly hamper patient’s quality of life. Importantly, besides inducing undesirable adverse effects, conventional taxane-containing chemotherapy only benefits to a minor fraction (15–20%) of primary breast tumors. It is therefore of utmost importance to identify biomarkers able to select with high confidence the patients who are at high risk to resist to chemotherapy. These predictive biomarkers represent a necessary step towards therapeutic de-escalation and future design of new targeted strategies for chemoresistant breast tumors [76].

In a transcriptomic analysis of three independent cohorts of breast cancer patients treated in neo-adjuvant settings with taxanes, 17 genes encoding microtubule-regulating proteins were found differentially expressed in chemoresistant breast tumors [58], and one of the most strongly deregulated genes was *MTUS1*. Interestingly, ATIP3 expression was significantly down-regulated in taxane-sensitive tumors achieving pathological complete response to chemotherapy [58], including in the BL1 subtype of TNBC [77]. Lymph node metastases were also significantly less frequent among low-ATIP3 expressing tumors following treatment with paclitaxel, compared with high-ATIP3 tumors [78]. These results were rather unexpected as ATIP3 deficiency is associated with increased microtubule dynamics [21], which is opposite to the microtubule-stabilizing effect of taxanes.

At the molecular level, ATIP3 depletion leads to increased accumulation of paclitaxel along the microtubule lattice [78], which accounts for higher sensitivity to low doses of chemotherapy. These results are consistent with in vitro findings that microtubule instability at the plus ends may improve Taxol binding to microtubules [79]. During mitosis, ATIP3 deficiency induces centrosome amplification and formation of multipolar spindles, which are sources of aneuploidy. In the presence of low doses of taxanes, these mitotic abnormalities accumulate above a tolerable level and promote massive cell death [80]. Thus, ATIP3 deficiency induces aneuploidy, which paradoxically sensitizes cancer cells to paclitaxel treatment. In line with these molecular data, human breast tumors expressing low *MTUS1* levels were shown to exhibit elevated aneuploidy and chromosome instability (Figure 1).

Thus, by increasing microtubule dynamics at growing plus ends, ATIP3 deficiency both favors increased paclitaxel binding to microtubules in interphase and promotes the formation of multipolar spindles in mitosis. The potent effects of taxane-based chemotherapy in ATIP3-deficient breast tumors likely arise from a combination of both mechanisms.

## 6. New ATIP3-Associated Emerging Targets for Breast Cancer Therapy?

In conclusion, studies conducted over the past ten years with the aim to depict the expression and function of ATIP3 in breast cancer have successfully pointed out this protein as a robust prognostic biomarker of patient survival and a strong predictive biomarker for resistance to taxane-based chemotherapy. Down-regulation of *MTUS1* is associated with tumor progression and poor outcome for the patients, suggesting that therapies designed to restore physiological levels of *MTUS1* transcripts may be valuable. Epigenetic mechanisms underlying *MTUS1* down-regulation in various cancers include promoter methylation and regulation of ATIP mRNA turnover and stability by RNA binding proteins, long non-coding RNA and microRNAs. Although still challenging, epigenetic-targeted therapeutic strategies are rapidly developing [81,82] and may open new avenues for restoring endogenous ATIP3 levels in ATIP3-deficient breast tumors. Epidrugs (compounds targeting epigenetic enzymes) and antagomirs (that block the effects of inhibitory microRNAs) [83] are being considered as promising future therapeutic options in cancer and progresses are being made to improve their delivery [84]. However, strategies for targeting ATIP3-deficiency in breast tumors still require better knowledge of epigenetic mechanisms involved in *MTUS1* gene alterations in cancer.

Recent studies have also unveiled interesting molecular mechanisms associated with ATIP3 intracellular effects, both in interphase and mitosis. ATIP3 contributes to cellular homeostasis by regulating microtubule dynamics and maintaining mitotic spindle integrity. The demonstration ATIP3 interacts with EB1 in the cytosol and with KIF2A/Aurora A at the spindle pole, provides clues to the design of molecular therapies targeting ATIP3-deficient tumors. Indeed, high throughput screening of small molecules (either synthetic compounds or small peptides), able to mimic or prevent ATIP3 participation into these molecular complexes, may represent an interesting strategy to restore ATIP3 intracellular functions in ATIP3-deficient tumors. Furthermore, ATIP3 deficiency in breast tumors has been associated with increased centrosome amplification and aneuploidy, which renders them more susceptible to taxane-based chemotherapy [58]. While aneuploidy is a recognized hallmark of aggressive cancer, it also appears as an Achille’s heel of tumors as far as taxane treatment is concerned. This paradigm shift opens new therapeutic strategies to exploit cancer vulnerability. Increasing cancer cell aneuploidy above a threshold using low doses of microtubule-stabilizing drugs may represent a novel way to induce tumor shrinkage [80]. Future in-depth characterization of ATIP3-associated intracellular mechanisms in normal and cancer cells are warranted to facilitate the design of new molecular therapies targeting a subpopulation of ATIP3-deficient breast cancer patients. With approximately 170,000 new cases of ATIP3-deficient TNBC tumors diagnosed annually worldwide, such an ATIP3-associated targeted approach will likely represent a major step forward to face a major public health problem. Other ATIP3-deficient solid tumors may additionally benefit from these therapeutic advances.

## Figures and Tables

**Figure 1 cells-10-01080-f001:**
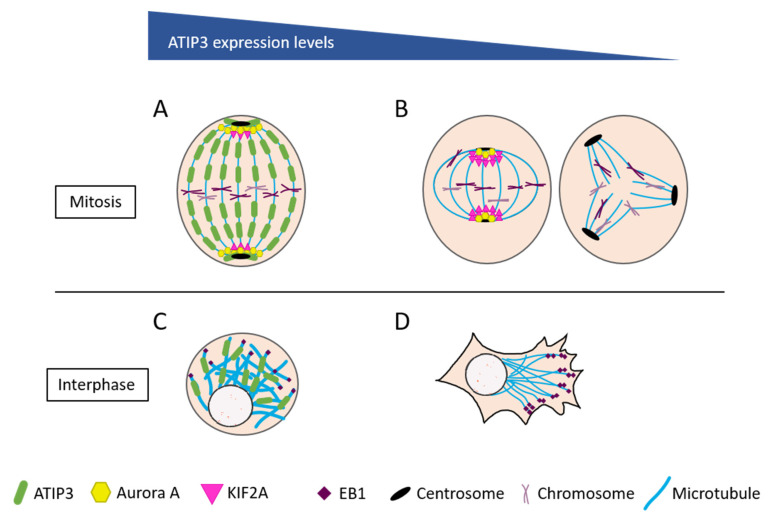
ATIP3-associated molecular mechanisms. (**A**) ATIP3 controls microtubule depolymerization by preventing KIF2A localization to the poles. (**B**) Left: ATIP3-deficient cells have a short metaphase spindle, due to increased KIF2A at the poles. Right: ATIP3-deficient cells show centrosome amplification and multipolar spindle formation, leading to aneuploidy. (**C**) ATIP3 stabilizes microtubules by negatively regulating EB1 turnover at microtubule plus-ends. (**D**) ATIP3-deficient cells are prone to increased directional migration and polarization, due to increased microtubule dynamics.

**Table 1 cells-10-01080-t001:** *MTUS1* gene status in human cancers.

Cancer Type	*MTUS1* Isoform	Detection Method	Expression Level	Prognosis *	Reference
Bladder	N.D.	IHC	Underexpressed	OS	[25]
	N.D.	RT-qPCR	Underexpressed	DFS	[24]
Breast	N.D.	Microarray	Underexpressed	N.D.	[23]
	ATIP3	Microarray		OS/MFS	[21]
	ATIP3	Microarray/IHC		N.D.	[19]
	ATIP3	Microarray/IHC		OS	[22]
Colorectal	N.D.	RNA-seq	Underexpressed	OS	[29]
	N.D.	IHC		N.D.	[27]
	N.D.	RT-qPCR/WB		N.D.	[26]
	N.D.	RT-qPCR		N.D.	[28]
Gallbladder	N.D.	Microarray/IHC	Underexpressed	DFS	[30]
Gastric	N.D.	RT-qPCR	Underexpressed	N.D.	[32]
Non small cell lung	N.D.	Microarray	Underexpressed	OS	[33]
Oral	N.D.	RT-qPCR	Underexpressed	N.D.	[38]
	ATIP3	IHC		OS	[36]
	N.D.	Microarray/IHC		OS	[34]
Prostate	ATIP1/ATIP3	RT-qPCR/IHC	Overexpressed	N.D.	[43]
Renal	N.D.	IHC	Underexpressed	N.D.	[41]
Uveal melanoma	N.D.	Microarray	Underexpressed	MFS	[42]

N.D. Not determined; OS: Overall survival, DFS: disease-free survival, MFS: metastasis free survival; IHC: immunohistochemistry, RT-qPCR: Reverse transcription quantitative polymerase chain reaction; RNAseq: RNA sequencing; WB: Western blot. * Underexpression of *MTUS1* is associated with reduced OS, MFS, and DFS.

**Table 2 cells-10-01080-t002:** ATIP3 localization and function in human and *Xenopus* cells.

	Human ATIP3	Reference	*Xenopus* ICIS	Reference
**Localization**	Microtubule Mitotic Spindle Centrosome	[19]	Centromere Centrosome	[55]
**Interacts with**	EB1 KIF2A DDA3	[49,51]	XKCM1 KIF2A AURKB INCENP TD-60	[55,56]
**Signaling**	ERK AURKA KIF2C CDC25B CDK1	[4,32,51,57]		
**Function**	Microtubule dynamics Spindle size Centrosome number Proliferation Migration Polarization EMT	[19,21,32,49,51,57,58]	Microtubule dynamics Mitotic spindle integrity	[55,56]

## Data Availability

Not applicable.

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
