# Peer review of "Microtubule-Associated Protein ATIP3, an Emerging Target for Personalized Medicine in Breast Cancer"

_cells, 2021, doi:10.3390/cells10051080_

Round 1
Reviewer 1 Report
The authors provides information regarding microtubule associated protein ATIP3 which is downregulated in cancer. This seems to be relatively novel protein which might have implications for various cancers.
can the author comment on in general what is known about microtubule associated proteins and cancer?
Can the authors show ATIP3 on figure 1? How does it interact with proteins shown. The authors might want to focus on ATIP3 only instead of focusing on ATIP1 initially.
Can you say how are these isoforms regulated in cancer? are they generally downregulated on in a isoform specific manner? It will be interesting to show the isoforms on a figure and show ATIP3s normal physiological roles with a table.
Author Response
We thank the reviewer for this positive comment on our manuscript. Please find the response in the attachment.

Reviewer 2 Report
This is a well-written paper reviewing the possible roles of ATIP3 in breast cancer and, actually, in other cancers as well. The authors have given a nice description of microtubule instability, which is not always the case in many papers I have reviewed. They have also included a very useful table showing that ATIP3 is under-expressed in many cancers, not just breast cancer, but also that it is over-expressed in prostate cancer. Clearly the story is a complex one and well worth investigating, so the authors are doing a service by publishing this review. Figure 1 gives a very good summary of the effect of lowering ATIP3 on a cell.
This said, the authors are on the horns of a dilemma. In a sense, they have a negative problem here. If there was a protein that was up-regulated in cancers then it would be useful to design a drug to inhibit the action of this protein, but what we have here is the opposite: a protein that is down- regulated in cancer. The authors speculate that one could take advantage of this by preventing the degradation of the mRNA for ATIP3 or preventing methylation of the promoter of the gene encoding the protein. They do not mention any specific methods for doing this, which, at this stage of development of the concept, is probably apropos, but they should address one or two of the following: 1. State that they do not know exactly how to up-regulate the protein and 2. Speculate on how to block methylation of one specific promoter without blocking methylation of other promoters and potentially wreaking havoc in normal cells. As I said, it is fine not to have the answers but the authors' contribution would be magnified if they shared more of their thinking. It would be of interest if the authors were to speculate as to why ATIP3 was up-regulated in prostate cancer.
There are also some wording issues, although not very many. Some spellcheck would be good. Here are some specific issues:
- Lines 31-32. The authors state that the MTSG1 product, which appears to be ATIP3, has anti-proliferative properties. What exactly are its anti-proliferative properties? If it is only that it is down-regulated in cancers that should be made clear. This is a negative characterization. We do not hear in the literature, to use a silly example, that hemoglobin has anti-phosphorylation properties---because it is not a kinase. Does ATIP3 actively inhibit proliferation? If so, how?
- Line 36. "seven transmembrane domain receptor" should be "seven-domain transmembrane receptor".
- Line 42. "were useful to demonstrate" should be either "were used to demonstrate" or "were useful for demonstrating". The two alternatives have slightly different nuances in their meanings. The authors should use whichever is closest to their intended meaning.
- Line 113. "EB1 autonomously binds a preferential, GTP-dependent, conformational site at the microtubule plus end". This sentence has no real meaning. How is EB1 binding "autonomous". How can a site be "preferential", "GTP-dependent" or "conformational"? The authors should make this very clear.
Author Response
We warmly thank the reviewer for his/her detailed and positive evaluation of our work. Please find the response in the attachment.
